# Distilling Visual Priors from Self-Supervised Learning

**Abstract.** Convolutional Neural Networks (CNNs) are prone to overfit small training datasets. We present a novel two-phase pipeline that leverages self-supervised learning and knowledge distillation to improve the generalization ability of CNN models for image classification under the data-deficient setting. The first phase is to learn a teacher model which possesses rich and generalizable visual representations via self-supervised learning, and the second phase is to distill the representations into a student model in a self-distillation manner, and meanwhile fine-tune the student model for the image classification task. We also propose a novel margin loss for the self-supervised contrastive learning proxy task to better learn the representation under the data-deficient scenario. Together with other tricks, we achieve competitive performance in the VIPriors image classification challenge.

**Keywords:** Self-supervised Learning, Knowledge-distillation

## 1 Introduction

Convolutional Neural Networks (CNNs) have achieved breakthroughs in image classification [8] via supervised training on large-scale datasets, e.g., ImageNet [4]. However, when the dataset is small, the over-parameterized CNNs tend to simply memorize the dataset and can not generalize well to unseen data. To alleviate this over-fitting problem, several regularization techniques have been proposed, such as Dropout [15], BatchNorm [11]. In addition, some works seek to combat with over-fitting by re-designing the CNN building blocks to endow the model with some encouraging properties (e.g., translation invariance [12]).

Recently, self-supervised learning has shown a great potential of learning useful representation from data without external label information. In particular, the contrastive learning methods [7, 1] have demonstrated advantages over other self-supervised learning methods in learning better transferable representations for downstream tasks. Compared to supervised learning, representations learned by self-supervised learning are unbiased to image labels, which can effectively prevent the model from over-fitting the patterns of any object category. Furthermore, the data augmentation in modern contrastive learning [1] typically involves diverse transformation strategies, which significantly differ from those used by supervised learning. This may also suggest that contrastive learning can better capture the diversity of the data than supervised learning.

In this paper, we go one step further by exploring the capability of contrastive learning under the data-deficient setting. Our key motivation lies in the realization that the label-unbiased and highly expressive representations learned by self-supervised learning can largely prevent the model from over-fitting the small training dataset. Specifically, we design a new two-phase pipeline for data-deficient image classification. The first phase is to utilize self-supervised contrastive learning as a proxy task for learning useful representations, which we regard as visual priors before using the image labels to train a model in a supervised manner. The second phase is use the weight obtained from the first phase as the start point, and leverage the label information to further fine-tune the model to perform classification.

In principle, self-supervised pre-training is an intuitive approach for preventing over-fitting when the labeled data are scarce, yet constructing the pre-training and fine-tuning pipeline properly is critical for good results. Specifically, there are two problems to be solved. First, the common practice in self-supervised learning is to obtain a memory bank for negative sampling. While MoCo [7] has demonstrated accuracy gains with increased bank size, the maximum bank size, however, is limited in the data-deficient setting. To address this issue, we propose a margin loss that can reduce the bank size while maintaining the same performance. We hope that this method can be helpful for fast experiments and evaluation. Second, directly fine-tuning the model on a small dataset still faces the risk of over-fitting, based on the observation that fine-tuning a linear classifier on top of the pre-train representation can yield a good result. We proposed to utilize a recent published feature distillation method [9] to perform self-distillation between the pre-trained teacher model and a student model. This self-distillation module plays a role of regularizing the model from forgetting the visual priors learned from the contrastive learning phase, and thus can further prevent the model from over-fitting on the small dataset.

## 2    Related Works

**Self-supervised learning**    focus on how to obtain good representations of data from heuristically designed proxy tasks, such as image colorization [21], tracking objects in videos [17], de-noising auto-encoders [16] and predicting image rotations [6]. Recent works using contrastive learning objectives [18] have achieved remarkable performance, among which MoCo [7, 2] is the first self-supervised method that outperforms supervised pre-training methods on multiple downstream tasks. In SimCLR [1], the authors show that the augmentation policy used by self-supervised method is quite different from the supervised methods, and is often harder. This phenomenon suggests that the self-supervised learned representations can be more rich and diverse than the supervised variants.

**Knowledge distillation**    aims to distill useful knowledge or representation from a teacher model to a student model [10]. Original knowledge distillation uses the predicted logits to transfer knowledge from teacher to student [10]. Then, some works found that transferring the knowledge conveyed by the feature map

from the teacher to student can lead to better performance [14, 20]. Heo *et al.* [9] provided a overhaul study of how to effectively distill knowledge from the feature map, which also inspires our design for knowledge distillation. Self-distillation uses the same model for both teacher and student [5], which has been shown to improve the performance of the model. We utilize the self-distillation method as a regulation term to prevent our model from over-fitting.

## 3   Method

Our method contains two phases, the first phase is to use the recently published MoCo v2 [2] to pre-train the model on the given dataset to obtain good representations. The learned representations can be considered as visual priors before using the label information. The second phase is to initialize both the teacher and student model used in the self-distillation process with the pre-trained weight. The weight of the teacher is frozen, and the student is updated using a combination of the classification loss and the overhaul-feature-distillation (OFD) [9] loss from the teacher. As a result, the student model is regularized by the representation from the teacher when performing the classification task. The two phases are visualized in Fig. 1.

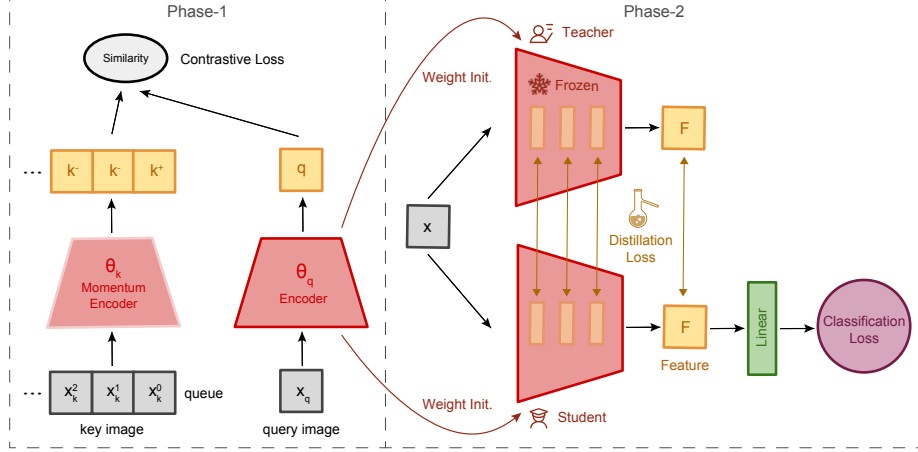

Fig. 1: The two phases of our proposed method. The first phase is to construct a useful visual prior with self-supervised contrastive learning, and the second phase is to perform self-distillation on the pre-trained checkpoint. The student model is fine-tuned with a distillation loss and a classification loss, while the teacher model is frozen.

### 3.1   Phase-1: Pre-Train with Self-Supervised Learning

The original loss used by MoCo is as follows:

$$\mathcal{L}_{\text{moco}} = - \log \left[ \frac{\exp\left(\mathbf{q} \cdot \mathbf{k}^+ / \tau\right)}{\exp\left(\mathbf{q} \cdot \mathbf{k}^+ / \tau\right) + \sum_{\mathbf{k}^-} \exp\left(\mathbf{q} \cdot \mathbf{k}^- / \tau\right)} \right], \tag{1}$$

where $\mathbf{q}$ and $\mathbf{k}^+$ is a positive pair (different views of the same image) sampled from the given dataset $\mathcal{D}$, and $\mathbf{k}^-$ are negative examples (different images). As shown in Fig. 1, MoCo uses a momentum encoder $\theta_k$ to encode all the $\mathbf{k}$ and put them in a queue for negative sampling, the momentum encoder is a momentum average of the encoder $\theta_q$:

$$\theta_k \leftarrow \eta \theta_k + (1 - \eta) \theta_q. \tag{2}$$

As shown in MoCo [7], the size of the negative sampling queue is crucial to the performance of the learned representation. In a data-deficient dataset, the maximum size of the queue is limited, we propose to add a margin to the original loss function to help the model obtain a larger margin between data samples thus help the model obtain a similar result with fewer negative examples.

$$\mathcal{L}_{\text{margin}} = - \log \left[ \frac{\exp\left(\left(\mathbf{q} \cdot \mathbf{k}^+ - m\right) / \tau\right)}{\exp\left(\left(\mathbf{q} \cdot \mathbf{k}^+ - m\right) / \tau\right) + \sum_{\mathbf{k}^-} \exp\left(\mathbf{q} \cdot \mathbf{k}^- / \tau\right)} \right]. \tag{3}$$

### 3.2   Phase-2: Self-Distill on Labeled Dataset

The self-supervised trained checkpoint from phase-1 is then used to initialize the teacher and student for fine-tuning on the whole dataset with labels. We choose to use OFD [9] to distill the visual priors from teacher to student. The distillation process can be seen as a regulation to prevent the student from over-fitting the small train dataset and give the student a more diversed representation for classification.

The distillation loss can be formulated as follows:

$$\mathcal{L}_{\text{distill}} = \sum_{\mathbf{F}} d_p \left( \text{StopGrad}\left(\mathbf{F}_t\right), r(\mathbf{F}_s) \right), \tag{4}$$

where $\mathbf{F}_t$ and $\mathbf{F}_s$ stands for the feature map of the teacher and student model respectively, the StopGrad means the weight of the teacher will not be updated by gradient descent, the $d_p$ stands for a distance metric, $r$ is a connector function to transform the feature from the student to the teacher.

Along with a cross-entropy loss for classification:

$$\mathcal{L}_{\text{ce}} = - \log p(y = i | \mathbf{x}), \tag{5}$$

the final loss function for the student model is:

$$\mathcal{L}_{\text{stu}} = \mathcal{L}_{\text{ce}} + \lambda \mathcal{L}_{\text{distill}}. \tag{6}$$

The student model is then used for evaluation.

# 4    Experiments

**Dataset**  Only the subset of the ImageNet [4] dataset given by the VIPrior challenge is used for our experiments, no external data or pre-trained checkpoint is used. The VIPrior challenge dataset contains 1,000 classes which is the same with the original ImageNet [4], and is split into train, val and test splits, each of the splits has 50 images for each class, resulting in a total of 150,000 images. For comparison, we use the train split to train the model and test the model on the validation split.

**Implementation Details**  For phase-1, we set the momentum $\eta$ as 0.999 in all the experiments as it yields better performance, and the size of the queue is set to 4,096. The margin $m$ in our proposed margin loss is set to be 0.6. We train the model for 800 epochs in phase-1, the initial learning rate is set to 0.03 and the learning rate is dropped by 10x at epoch 120 and epoch 160. Other hyperparameter is set to be the same with MoCo v2 [2],

For phase-2, the $\lambda$ in Eq. 6 is set to $10^{-4}$. We also choose to use $\ell_2$ distance as the distance metric $d_p$ in Eq. 4. We train the model for 100 epochs in phase-2, the initial learning rate is set to 0.1 and is dropped by 10x every 30 epochs.

**Ablation Results**  We first present the overall performance of our proposed two phase pipeline, then show some ablation results.

As shown in Tab. 1, supervised training of ResNet50 [8] would lead to over-fitting on the train split, thus the validation top-1 accuracy is low. By first pre-training the model with the phase-1 of our pipeline, and fine-tuning a linear classifier on top of the obtained feature representation [18], we can reach a 6.6 performance gain in top-1 accuracy. This indicates that the feature learned from self-supervised learning contain more information and can generalize well on the validation set. We also show that fine-tuning the full model from phase-1 can reach better performance compared to only fine-tuning a linear classifier, which indicates that the weight from phase-1 can also serve as a good initialization, but the supervised training process may still cause the model to suffer from over-fitting. Finally, by combining phase-1 and phase-2 together, our proposed pipeline achieves 16.7 performance gain in top-1 accuracy over the supervised baseline.

**The effect of our margin loss**  Tab. 2 shows that effect of the number neg-ative samples in contrastive learning loss, the original loss function used by MoCo v2 [7] is sensitive to the number of negatives, the fewer negative, the lower the linear classification result is. Our modified margin loss can help alle-viate the issue with a margin to help the model learn a larger margin between data points. The experiments show that our margin loss is less sensitive to the number negatives and can be used in a data-deficient setting.

| ResNet50 | #Pretrain Epoch | #Finetune Epoch | Val Acc |
|---|---|---|---|
| Supervised Training | - | 100 | 27.9 |
| Phase-1 + finetune fc | 800 | 100 | 34.5 |
| Phase-1 + finetune | 800 | 100 | 39.4 |
| Phase-1 + Phase-2 (Ours) | 800 | 100 | 44.6 |

Table 1: Training and Pre-training the model on the train split and evaluate the performance on the validation split on the given dataset. 'finetune fc' stands for train a linear classifier on top of the pretrained representation, 'finetune' stands for train the weight of the whole model. Our proposed pipeline (Phase-1 + Phase-2) can have 16.7 performance gain in top-1 validation accuracy.

|  | #Neg | Margin | Val Acc |
|---|---|---|---|
| | 4096 | - | 34.5 |
| MoCo v2 [7] | 1024 | - | 32.1 |
| | 256 | - | 29.1 |
| | 4096 | 0.4 | 34.6 |
| Margin loss | 1024 | 0.4 | 34.2 |
| | 256 | 0.4 | 33.7 |

Table 2: The Val Acc means the linear classification accuracy obtained by fine-tune a linear classifier on top of the learned representation. The original MoCo v2 is sensitive to the number of negative, the performance drops drastically when number negatives is small. Our modified margin loss is less sensitive to the number negatives, as shown in the table, even has 16x less negatives the performance only drops 0.9.

|  | #Pretrain Epoch | #Finetune Epoch | Test Acc |
|---|---|---|---|
| Phase-1 + Phase-2 | 800 | 100 | 47.2 |
| +input Resolution 448 | 800 | 100 | 54.8 |
| +ResNeXt101 [19] | 800 | 100 | 62.3 |
| +label-smooth [13] | 800 | 100 | 64.2 |
| +Auto-Aug [3] | 800 | 100 | 65.7 |
| +TenCrop | 800 | 100 | 66.2 |
| +Ensemble two models | 800 | 100 | 68.8 |

Table 3: The tricks used in the competition, our final accuracy is 68.8 which is a competitive result in the challenge. Our code will be made public. Results in this table are obtain by train the model on the combination of train and validation splits.

**Competition Tricks** For better performance in the competition, we combine the train and val split to train the model that generate the submission. Several other tricks and stronger backbone models are used for better performance, such as Auto-Augment [3], ResNeXt [19], label-smooth [13], TenCrop and model ensemble. Detailed tricks are listed in Tab. 3.

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
