# OpenReview forum: "Distilling Visual Priors from Self-Supervised Learning"
_thecvf.com/ECCV/2020/Workshop/VIPriors — VIPriors Poster_

### Official Review · AnonReviewer1 · 2020-07-23
**Interesting method, well-written paper**

**Confidence:** 4
**Rating:** 8

**Review:**

#### 1. [Summary] In 2-3 sentences, describe the key ideas, experiments, and their significance.
The paper proposes to use contrastive pre-training to construct a visal prior (i.e. the model weights) and subsequently initializes both a teacher and student model with the pre-trained weights to finetune the student network on the dataset while  imposing an additional distillation loss between the frozen teacher and unfrozen student networks.

#### 2. [Strengths] What are the strengths of the paper? Clearly explain why these aspects of the paper are valuable.
* The paper is clear and is easy to understand
* The method is interesting and seems to perform well
* The ablation studies clealy show which portion of the performance gain can be accredited to the proposed method and which portion is due to additional tricks, such as data augmentation.

#### 3. [Weaknesses] What are the weaknesses of the paper? Clearly explain why these aspects of the paper are weak.
* It would have been nice to include experiments on an additional (toy) dataset such as MNIST or SVHN to show that the method generalizes to other tasks.
* Conclusion section is missing; the paper is ending rather abruptly.

#### 4. [Overall rating] Paper rating
8: Top 50% of accepted papers, clear accept

#### 5. [Justification of rating] Please explain how the strengths and weaknesses aforementioned were weighed in for the rating.
The method is interesting and seems to perform well. In addition, the paper is well-written. Please consider extending the paper with a "Conclusion" section.

#### 6. [Detailed comments] Additional comments regarding the paper (e.g. typos or other possible improvements you would like to see for the camera-ready version of the paper, if any.)
* What is $\tau$ in equation (1)?

---

### Official Review · AnonReviewer2 · 2020-07-27
**Distilling Visual Priors from Self-Supervised Learning**

**Confidence:** 5
**Rating:** 7

**Review:**

#### 1. [Summary] In 2-3 sentences, describe the key ideas, experiments, and their significance.
The method has two stages: (i) a teacher network is trained with contrastive learning to obtain feature representation, (ii) the knowledge of the teacher network is transferred to student network by distillation, in the meantime, the student network is also finetuned with labels.

#### 2. [Strengths] What are the strengths of the paper? Clearly explain why these aspects of the paper are valuable.
- 2-stage method
- Using a margin to overcome the small bank size problem
#### 3. [Weaknesses] What are the weaknesses of the paper? Clearly explain why these aspects of the paper are weak.
- No conclusion and discussion section.


#### 4. [Overall rating] Paper rating
7

#### 5. [Justification of rating] Please explain how the strengths and weaknesses aforementioned were weighed in for the rating.


#### 6. [Detailed comments] Additional comments regarding the paper (e.g. typos or other possible improvements you would like to see for the camera-ready version of the paper, if any.)

- How is the margin value chosen? In the text, it is given as 0.6 but in the table, the value is 0.4.
- Related works: Why is there any margin loss part?
- Missing citations:
	> L.29:".. simply memorize the dataset and can not generalize well to unseen data.."
	> L.31:"..some works.." (only given one)

---

### Decision · Program_Chairs · 2020-07-29

**Decision:**

Accept (Poster)

**Comment:**

It is our pleasure to inform you that your paper has been accepted to the poster track of 1st Visual Inductive Priors for Data-Efficient Deep Learning Workshop.

Please note the following deadlines:
* August 11, 2020 - workshop material, including:
 * paper in PDF format;
 * pre-recorded video presentation;
 * slides of the presentation in PDF.
* September 15, 2020 - camera-ready paper

The reviews can be found on OpenReview. Please take these comments and suggestions into account when preparing the camera-ready version of your paper, which is due September 15, 2020. The camera-ready paper should be uploaded to OpenReview.

As part of the workshop, each accepted paper must submit a pre-recorded 90 second talk before August 11, 2020. You will receive more information on how to upload the material shortly. The requirements for the video are:
* Duration: maximum 90 seconds
* MP4 format
* File size max. 100 MB
* Has an inset with a video of the speaker
* 16:9 aspect ratio (strongly preferred)
* 1920x1080 resolution (strongly preferred, at least 720 height)

Our suggested software for pre-recording your presentation is Zoom. For more information, please refer to the following guides:
How to record with Zoom Guide: http://homepages.inf.ed.ac.uk/rbf/ECCV2020HowtoRecordusingZoom.pdf
How to Record with Zoom tutorial: https://www.youtube.com/watch?v=CR199W7HdC0
Please ensure that at least one of the authors of the paper is available to attend the workshop during the allotted times. Note that the workshop will take place in two sessions spread across time zones (details are to follow). We will send instructions on how to connect to the workshop as soon as possible. The schedule for all talks and papers will be posted soon at the workshop website: https://vipriors.github.io.

We look forward to seeing you at the workshop!